# Association of Dental Caries, Retained Roots, and Missing Teeth with Physical Status, Diabetes Mellitus and Hypertension in Women of the Reproductive Age

**DOI:** 10.3390/ijerph16142565

**Published:** 2019-07-18

**Authors:** Najla Dar-Odeh, Sary Borzangy, Hamzah Babkair, Lamis Farghal, Ghufran Shahin, Sawsan Fadhlalmawla, Walaa Alhazmi, Sarah Taher, Osama Abu-Hammad

**Affiliations:** 1College of Dentistry, Taibah University, Al Madinah Al Munawara 43353, Saudi Arabia; 2School of Dentistry, University of Jordan, Amman 11942, Jordan; 3Private Sector, Al Madinah Al Munawara 42313, Saudi Arabia; 4Department of Periodontology, Faculty of Dentistry, King Abdulaziz University, Jeddah 21589, Saudi Arabia

**Keywords:** Dental caries, Retained roots, Missing teeth, Hypertension, Diabetes mellitus, Prediction

## Abstract

Objectives: To investigate in women of reproductive age a possible association between particular dental diseases—dental caries, retained roots, and missing teeth—with some systemic conditions—physical status score- ASA (American Society for Anesthesiologists), diabetes mellitus, and hypertension. Methods: Dental and medical history were retrieved from the electronic files of dental patients. Statistical analysis was performed using cross tabulation with the Chi-square test to explore the significance of an association between variables pertaining to dental diseases and the investigated systemic conditions. Logistic regression was further used to explore the significance of the above dental diseases as predictors for systemic conditions. Results: A total of 1768 female patients in the age range 18–55 were included, with a mean age of 31.2 ± 10.13 years. A total of 228 (12.9%) patients had a chronic systemic disease within the ASA II category, 66 (3.7%) were diabetic, and 76 (4.3%) were hypertensive. Missing teeth were significantly associated with the ASA II category, diabetes mellitus, and hypertension (*p* < 0.001, *p* = 0.009, *p* = 0.005 respectively), while retained roots were significantly associated with the ASA II category only (*p* = 0.023). Logistic regression showed a low predictive capacity of models describing the three systemic conditions. Conclusions: Diabetes mellitus and hypertension were the most common systemic diseases among the study sample. While carious teeth had no significant association with the investigated systemic conditions, retained roots were significantly associated with the ASA II category only, and missing teeth were significantly associated with all investigated systemic conditions. However, oral diseases expressed a low predictive power of these systemic conditions.

## 1. Introduction

Hypertension (HTN), diabetes mellitus (DM), and other cardiovascular risk factors were shown to be highly prevalent among women in developing countries [1]. It is estimated that HTN and DM affect 20% and 25% of women, respectively [1]. It is also well established that there is a parallel increase in the prevalence of dental diseases, such as caries, retained roots, and periapical jaw lesions, in these populations [2]. Recent studies reported that among women there is a high prevalence of retained roots and periapical jaw lesions, reaching 25% and 50%, respectively [3]. Such oral diseases are often associated with certain lifestyle factors that adversely influence oral and general health [4,5]. Their impact on general health, compounded by the associated high financial and social burdens, necessitates increased attention from policy and decision makers [6]. The influence of dental diseases on general health is mostly attributed to the nature of these diseases, which are primarily chronic bacterial infections that have a local effect of compromised tooth support and a generalized effect, whereby the increased systemic levels of inflammatory mediators contribute to endothelial dysfunction and carotid artery plaque formation [7,8].

Numerous studies have been conducted on the periodontitis–systemic disease inter-relationship [9]. Furthermore, within the context of reproductive women, cardiovascular diseases and adverse pregnancy outcomes, such as low birth weight and preterm birth, have been linked to periodontitis [10]. However, investigating the correlation between dental diseases in the form of carious teeth, retained roots, and missing teeth on one hand and systemic disease on the other hand has received less attention.

Earlier studies found a correlation between tooth loss and HTN in males but not females [11]. More recent studies focused on the positive association between tooth loss and HTN in post-menopausal women [12,13]. Furthermore, it was found that DM and HTN were independently associated with a higher experience of missing teeth [14]. 

It is notable, though, that previous research was concerned with gender differences or with postmenopausal women. More data is needed on the association between poor oral health in the form of dental caries, retained roots, and missing teeth with the physical status of younger women or women of the reproductive age. 

This study was conducted with a large sample of female dental patients of reproductive age to estimate the prevalence and types of systemic diseases they had, and to investigate whether specific dental diseases (carious teeth, missing teeth, and retained roots) are predictive for their physical status.

## 2. Methods

This descriptive retrospective study was approved by the Taibah University College of Dentistry Research Ethics Committee, reference number (TUCDREC/20171213/Dar-Odeh). 

The study was carried out at Taibah University Dental College and Hospital (TUDCH) in Al Madinah Al Munawarah, western Saudi Arabia. It involved retrieving electronic files stored on the clinical software system used at TUDCH (CS R4 Clinical + Practice Management Software, Carestream Dental Ltd., Rochester, NY, USA) during the period 1 January 2018 to 31 December 2018. Data retrieved from the electronic files were obtained from patients during their initial (screening) visit to the hospital. All patients who gave a history of DM and/or HTN, or were suspected to have these diseases, were further tested to confirm their physical fitness to receive dental treatment. All adult patients of reproductive age (18–55 years) were included. Extracted data included patients’ age, number of carious teeth, number of retained roots, number of missing teeth, and systemic diseases.

### Statistical Analysis

Descriptive statistics were generated for variables pertinent to dental diseases—in the form of retained roots, and carious and missing teeth—and general health variables—ASA II physical status score, DM, and HTN. Cross tabulation with Pearson Chi-square was used to analyze the significance of the association between both types of variables. Logistic regression was used to explore the significance of age and dental disease variables, as predictor variables for the occurrence of general health variables. For each significant predictor, the calculation of regression coefficients and 95% CIs were obtained. Statistical significance was set at *p* < 0.05.

For cross tabulation, patients were grouped in two groups for each dental disease or systemic disease (i.e., present or absent) and patients were grouped in two age categories, whereas for logistic regression, scale data for each patient was used (i.e., actual numbers of missing teeth, retained roots and carious teeth). Age was also expressed in years.

Data were analyzed using the statistical analysis software (SPSS version 21 (IBM Corp., Armonk, NY, USA).

## 3. Results

A total of 1768 female patients (mean age: 31.2 ± 10.13 years and age range: 18–55 years) visited the dental hospital during 2018. The average number of carious teeth for patients was 5.29 ± 4.10 teeth, with a range of 0–22 carious teeth. The average number of retained roots was 0.825 ± 1.81 teeth, with a range of 0–21 teeth. The average number of missing teeth was 2.60 ± 3.82 teeth, with a range of 0–32 teeth. A total of 225 (12.7%) patients had a chronic systemic disease that is classified as ASA II according to the classification of the American Society of Anesthesiologists (Figure 1). A description of the systemic diseases in the study sample is shown in Table 1.

When the variables of physical status score, DM, and HTN were cross tabulated with dental diseases, it was found that the number of carious teeth was not significantly associated with any of them, however, presence of retained roots was significantly associated with ASA II (*p* < 0.05) (Table 1). Missing teeth were significantly associated with ASA II, DM, and HTN (Table 1).

The logistic regression indicated that age and retained roots were significant predictors for ASA category, while age and missing teeth were significant predictors for HTN and DM (Table 2).

However, the numbers of subjects who were healthy and free of these systemic conditions were significantly higher than those who were affected by them. The odds ratios for patients who were either ASA II, diabetic, or hypertensive were found to be as low as 14.8%, 3.9%, and 4.5%, respectively. This indicates significantly lower number of patients with positive systemic disease findings.

The predictions in the three models were highly accurate for patients not having the diseases rather than for patients affected by them. For the three models predicting ASA II, DM, and HTN, the Nagelkerke R^2^ values were 0.110, 0.066, and 0.187 respectively.

The differences between observed and predicted values by the model (Hosmer and Lemeshow test) indicated an acceptable predictive power for the diabetes model only. The *p* values for the three logistic models of ASA II, diabetes, and hypertension were 0.005, 0.068, and 0.022, respectively.

## 4. Discussion

This study was part of a larger study initiated a few years ago, aiming to investigate and promote the oral health status of women in Al Madinah, particularly considering that a substantial proportion were faced with a number of barriers against obtaining appropriate oral health care [15,16], and this may be paralleled by obstacles in obtaining appropriate medical care. This study investigated a potential link between some common dental diseases and other common cardiovascular risk factors in women of reproductive age. These diseases were included due to their relatively high prevalence among the study sample. The study sample was women of reproductive age, a critical, relatively long period in a woman’s life. Medical history documented in the electronic files was obtained from patients during their initial screening visit. All patients who gave a history of HTN and/or DM, or were suspected to have these diseases, were further tested to insure their fitness for dental treatment. However, we could not rule out that some of them may have had undiagnosed diseases, and subsequently the prevalence of reported systemic diseases may actually be higher. In this group of women, the existing medical and dental history may be complicated by other confounders that worsen the status of oral healthcare. It is well-established now that many of these women are basically immigrants and have a language barrier complicated by low education, causing many of them to be illiterate in Arabic and/or English, and hence incapable of communicating or relaying their fears and expectations to their care givers [10]. They may also have financial barriers [17] and transportation problems [2], making access to healthcare limited and jeopardized. The findings of this study confirm the high prevalence of dental caries, as 86–89% (according to the age group) of women had carious teeth. Missing teeth were also prevalent, especially in the older age group (31–55 years). Despite the fact that the study sample was not in the old age group, approximately one in three patients had one or more retained root. Retained roots represent an advanced stage of dental caries, and highlight negligent behavior in seeking oral healthcare. It was not the aim of this study to investigate radiographic jaw lesions like periapical lesions, but we think that due to detected lesions of caries and retained roots, many of those women had associated bone lesions, like abscesses, cysts, and granulomas, the expected sequelae to retained roots. A previous study found a prevalence of 53.6% of periapical lesions among females in this geographic area [18]. Within the study sample, the number of retained roots were significantly associated with older age. This may be explained by a previous history of caries and as patients grew older, they just neglected to seek treatment for these teeth. Carious teeth gradually transformed into retained roots as patients neglected to seek dental extraction and subsequent prosthesis [19]. Retained roots were also significantly associated with (and significantly predicted) ASA II category. This may be explained by two main points. First, ASA II patients were generally older, and second, retained roots seem to be part of the general health condition that is worse in the ASA II category patients. 

All medically compromised patients in this study were in the ASA II category, as the facilities in the dental hospital do not allow for providing oral health care for patients with more complicated medical histories. Among these patients, HTN was the most commonly reported systemic disease, followed by DM. This confirms that these diseases are actually prevalent in females in Saudi Arabia [1]. Like retained roots, missing teeth were significantly associated with older age and ASA II category. Furthermore, missing teeth were additionally associated with DM and HTN. Older patients are expected to experience tooth loss more commonly than the younger age group, however, it is unfortunate for these women, since they are ≤55 years of age. 

The pseudo R^2^ results in logistic regression, indicate a small variance of the dependent variables as an effect of the independent variables in the three models, indicating the low predictive power of the regression model. Although the percentages of correct predictions for the three models were 87.0% for ASA, 96.3% for diabetes, and 95.7% for hypertension, it appears that predictions for the absence of the systemic disease or condition were much higher and more accurate than predictions of the presence of the condition. Besides this, the number of patients affected by all these systemic conditions is significantly lower than those unaffected by them.

Missing teeth were a statistically significant predictor for ASA II category, DM, and HTN. However, only 6.6% of the variance of the predicted values for patients’ condition regarding diabetes can be explained based on the predictor variables (based on the pseudo R^2^ values). It is important to note that patients are more susceptible to tooth loss due to their susceptibility to periodontal diseases associated with prolonged hyperglycemia and poor glycemic control; consequently, compromised tooth support will lead to tooth loss if patients do not receive dental attention at the appropriate time. This mechanism can explain the hidden association between tooth loss and diabetes. Consequences to tooth loss are not only evident in the loss of oral functions and impaired quality of life, but also psychological disturbances are anticipated. A recent study found that among people aged 65 and over, the severity of depression increases with a higher number of missing teeth and the number of decayed teeth [20], an association that was detected previously between DM and depression [21].

Recent studies also confirmed the relationship between HTN and periodontitis, the latter being typically known as a precursor of tooth loss. A recent systematic review suggested that periodontitis is associated with a higher risk of HTN [22]. It was thought previously that complete, but not partial, tooth loss is associated with adverse dietary behavior consisting of a soft unhealthy diet, and hence a subsequent increased risk of cardiovascular disease [23]. However more recent studies confirmed that partial tooth loss was also associated with a higher risk of HTN, and this was explained by the infection–inflammation pathway related to periodontal disease [24]. 

Previous studies among female dental patients in this geographic area found a high prevalence of alveolar bone loss [18], which is mainly caused by periodontal disease. So, this link cannot be over-looked. The periodontal disease–low socio-demographic status–hypertension association was supported by recent studies [25]. In a recent large Korean study on adult females, hypertension showed a significant association with periodontitis, and this association was marked in females aged 30–59 years, while the strength of the association was highest in females aged 30–39 years and decreased with increasing age [26]. The same study found that the high-risk groups for this link were in the lower middle income quartile [26]. Insufficient masticatory function, poor oral hygiene, and oral inflammation were also associated with hypertension in subjects <65 years of age [27]. 

Dental caries in this study were not significantly associated with diabetes, hypertension, or even ASA II category. The lack of association may reflect the limited pathological effects of caries, which are still confined within the boundaries of tooth structure, with no spread to surrounding bone and consequently no systemic spread of inflammatory mediators would result. However, a recent Brazilian study reported an association between caries and hypertension [28]. It has been suggested that women show more dramatic hormonal changes due to reproductive factors during their lifetime [29]. Modifiable social factors, like low educational level, occupation, and income, largely contribute to unhealthy lifestyle behaviors and social disparities, and thus are related to a higher risk of HTN and DM, particularly in women [30].

The limitations of this study are attributed to its design. It was a retrospective analysis of patients’ clinical records, utilizing the data that were already present. There was no data on oral hygiene practices or socio-demographic factors that may influence oral health. Another limitation was the cross-sectional nature of the study, hence, it was impossible to know the true causal order of disease. While the dental diseases (caries, missing teeth, and retained roots) included in this study were based on clinical examination, medical history was self-reported by patients. There was no attempt made to obtain further details on the medical condition of participants, so there is a possibility that these patients had a more complicated medical history. There is also a possibility that the prevalence of HTN and DM is higher than is already documented. 

It can be concluded that: ASA II physical status score, HTN, and DM were significantly associated with missing teeth, while ASA II physical status score alone was significantly associated with retained roots. Oral findings displayed a low predictive power of the investigated systemic conditions. We recommend investigating other factors that may have direct relationship with these conditions.

## Figures and Tables

**Figure 1 ijerph-16-02565-f001:**
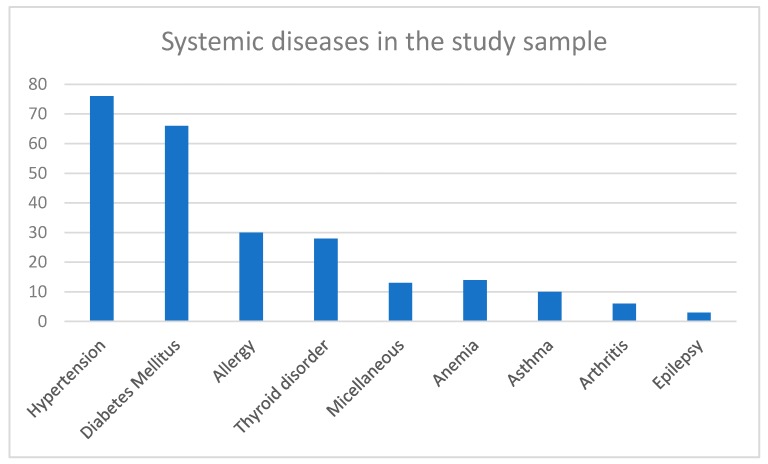
Frequency of systemic diseases in the study sample.

**Table 1 ijerph-16-02565-t001:** Comparison between patient groups with dental diseases (caries, retained roots -RR, and missing teeth) in terms of age, physical status score, hypertension (HTN), and diabetes mellitus (DM).

Dental Disease	Age Groups		Physical Status Score		DM		HTN	
	18–30 years N = 948 (53.6%)	31–55 years N = 820 (46.4%)	ASA I N = 1540	ASA II N = 228	No N = 1702	Yes N = 66	No N = 1692	Yes N = 76
Caries/No Carious Teeth	103 (10.9%) 845 (89.1%)	109 (13.3%) 711 (86.7%)	189 (12.3%) 1351 (87.7%)	23 (10.1%) 205 (89.9%)	208 (12.2%) 1494 (87.8%)	4 (6.1%) 62 (93.9%)	207 (12.2%) 1485 (87.8%)	5 (6.6%) 71 (93.4%)
*p* value	0.117		0.343		0.131		0.138	
RR/No RR	687(72.5%) 261 (27.5%)	507 (61.8%) 313 (38.2%)	1055 (68.5%) 485 (31.5%)	139 (61.0%) 89 (39.0%)	1155 (67.9%) 547 (32.1%)	39 (59.1%) 27 (40.9%)	1149 (67.9%) 543 (32.1%)	45 (59.2%) 31 (40.8%)
*p* value	<0.001		0.023		0.135		0.113	
Missing Teeth/No Missing Teeth	456 (48.1%) 492 (51.9%)	156 (19.0%) 664 (81.0%)	557 (36.2%) 983 (63.8%)	55 (24.1%) 173 (75.9%)	599 (35.2%) 1103 (64.8%)	13 (19.7%) 53 (80.3%)	597 (35.3%) 1095 (64.7%)	15 (19.7%) 61 (80.3%)
*p* value	<0.001		<0.001		0.009		0.005	

**Table 2 ijerph-16-02565-t002:** Models of logistic regression analysis of effect of oral conditions on physical status score, HTN, and DM.

Dependent Variable	Predictors	B	SE	Wald	*p* Value	Exp (B)
ASA Nagelkerke R^2^ = 0.110	Age	0.064	0.008	69.716	0.000	1.066
	Carious teeth	0.013	0.019	0.470	0.493	1.013
	RR	0.075	0.033	5.029	0.025	1.077
	Missing teeth	0.016	0.017	0.894	0.345	1.016
	Constant	−4.276	0.3026	200.069	0.000	0.014
DM Nagelkerke R^2^ = 0.187	Age	0.053	0.013	15.932	0.000	1.054
	Carious teeth	0.032	0.032	0.952	0.329	1.032
	RR	0.063	0.053	1.381	0.240	1.065
	Missing teeth	0.046	0.026	3.200	0.074	1.047
	Constant	−5.445	0.533	104.497	0.000	0.004
HTN Nagelkerke R^2^ = 0.066	Age	0.107	0.014	62.783	0.000	1.113
	Carious teeth	0.065	0.032	4.239	0.040	1.067
	RR	0.042	0.053	0.618	0.432	1.043
	Missing teeth	0.040	0.023	2.913	0.088	1.041
	Constant	−7.587	0.613	153.208	0.000	0.001

B: unstandardized beta weights; SE: standard error; Wald: Wald statistic; Exp (B): exponentiated B value or odds ratio; ASA: physical status score (American Society of Anesthesiologists), RR: retained roots. HTN: hypertension.

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
