# Peer review of "Association of Dental Caries, Retained Roots, and Missing Teeth with Physical Status, Diabetes Mellitus and Hypertension in Women of the Reproductive Age"

_ijerph, 2019, doi:10.3390/ijerph16142565_

Reviewer 1 Report

The article undoubtedly brings an added value to dental and medical public health area, however the conclusions which have been drawn by authors are 'too definite' and do not seem to reflect a real associations between presented and assessed variable and common medical health conditions.

The main value of article is a large number of cases included, but it does not preclude from the fact that results obtained were correctly interpreted. The term 'predictors' used in the article numerous of times is highly misleading, obtained data extrapolation leads to potential research bias. It could be equally true that diabetes, ASA status and hypertension, vastly impacting a quality of life, may be a primary reasons of not adequate oral health, not a subsequent consequences of high poor oral health.

The main title reveals a misconception of the primary idea of the study conducted.

Author Response

Reviewer number 1:

The honorable reviewer number 1 indicated the following:

The article undoubtedly brings an added value to dental and medical public health area, however the conclusions which have been drawn by authors are 'too definite' and do not seem to reflect a real associations between presented and assessed variable and common medical health conditions. 

The main value of article is a large number of cases included, but it does not preclude from the fact that results obtained were correctly interpreted. The term 'predictors' used in the article numerous of times is highly misleading, obtained data extrapolation leads to potential research bias. It could be equally true that diabetes, ASA status and hypertension, vastly impacting a quality of life, may be a primary reasons of not adequate oral health, not a subsequent consequences of high poor oral health. 

The main title reveals a misconception of the primary idea of the study conducted.

The honorable reviewers expressed concern about the interpretation of the results.

He/she is absolutely correct.  After the consultation with statistician, we had to replace linear regression with logistic regression.  The carried out statistics provided lower predictive value of the model with lower ‘pseudo’ R2 values as indicated by the Hosmer and Lemeshow test which is consistent with the honorable reviewer’s comments.  The honorable reviewer also indicated possible misinterpretation of results in terms of the presence of uninvestigated variables affecting general health.  This is also possible and has been introduced into the text throughout the article: in abstract, in materials and methods, in results, discussion and conclusions. Even we changed the title accordingly.  Changes are highlighted in yellow.

Also the honorable reviewer demanded the following:

Methods adequately described:  can be improved

Does introduction provide sufficient background and include relevant references: must be improved.

Is the research design appropriate: must be improved.

Are results clearly presented: must be improved.

Are conclusions supported by the results: must be improved.

Regarding the inclusion of more references in the introduction, unfortunately, we could not find any more references to address this point.

However, we have made significant changes to the abstract, methods section, results and conclusions as shown in highlighted text as requested.

Reviewer 2 Report

Dear authors,

Congratulations for the paper, correlating dental diseases with medical diseases.

All the best.

Author Response

Many thanks and appreciation for your opinion.

Reviewer 3 Report

Some methodological points should be better explained prior to publication:

1)      Why were all the variables categorized in two groups (page 2, line 74-76)? Statistical models can contain quantitative variables without losing information.

2)      It is clear from table 2 (R^2 values) that the multivariate models explain only a little part of the predictor variability. Authors should clearly explain the variables included in the models, In my opinion, other variables that could explain ASA score, HTN and DM were not mentioned.

3)      HTN and DM are categorical variables. Therefore, logistic regression despite of multiple linear regression should be used.

Author Response

Yes

Can   be improved

Must   be improved

Not   applicable

Does   the introduction provide sufficient background and include all relevant   references?

(x)

( )

( )

( )

Is   the research design appropriate?

( )

(x)

( )

( )

Are   the methods adequately described?

( )

( )

(x)

( )

Are   the results clearly presented?

( )

(x)

( )

( )

Are   the conclusions supported by the results?

( )

( )

(x)

( )

We have made significant modifications to the abstract section, methods, results and discussion and conclusions as shown in highlighted text.  We hope we have addressed your above points.

Comments and Suggestions for Authors

Some methodological points should be better explained prior to publication:

1)      Why were all the variables categorized in two groups (page 2, line 74-76)? Statistical models can contain quantitative variables without losing information.

2)      It is clear from table 2 (R^2 values) that the multivariate models explain only a little part of the predictor variability. Authors should clearly explain the variables included in the models, In my opinion, other variables that could explain ASA score, HTN and DM were not mentioned.

3)      HTN and DM are categorical variables. Therefore, logistic regression despite of multiple linear regression should be used.

Regarding point 1: we made the categorization only for cross tabulation purposes.  When the regression model was built, original data (scale data) for age, numbers of missing teeth, carious teeth and remaining roots were used. This has now been clarified in the materials and methods section.

Regarding point 2: you are absolutely right, thank you.  We have now pointed out this fact and that the regression model has low predictive power.  We pointed out to this in abstract, results, discussion and conclusions.

Regarding point 3: absolutely ri

Round  2

Reviewer 1 Report

Substantial efforts have been made to improve the content and reduce the risk of scientific bias. The altered title and the abstract clearly reflect the main idea of the study.

The final conclusions ought to be more 'neutral'.

Author Response

The comment was addressed and conclusions were modified in the last paragraph of discussion.

Reviewer 3 Report

acceptable

This manuscript is a resubmission of an earlier submission. The following is a list of the peer review reports and author responses from that submission.

Round  1

Reviewer 1 Report

The novelty of this paper is questionable, and the selection of methodology is controversial from the potential bias point of view.

The potential link between the described variables should be further investigated, including RCT's and longitudinal studies, also the properly chosen statistical  tests.

Reviewer 2 Report

Congratulations for the study but some revision need to be done.

Page 4. Table 1. Legend should be improved.

Line 12, page 5- Rephrase the paragraph.

Page 5. Table 2- Legend should be improved.

Page 7. Line 99. Add: None.

Reviewer 3 Report

This is a retrospective study which use medical history obtained from female patients during their initial screening in a hospital. Statistical analysis was performed to understand the relation between some of the medical diseases and dental diseases. Some dental diseases correlated with some of the medical diseases. The authors concluded that some dental diseases are predictors for physical status and some medical diseases.

The data of medical diseases are solely based on the information given by the patients during history taking. Diagnosis of the medical diseases was not performed. Besides authors stated that the patients had a language barrier. There is no socio-demographic data, economic condition of the patients which may greatly influence the occurrence of dental diseases. Although the authors acknowledged these limitations, without proper understanding of these factors, it is difficult to come out to the conclusion that has been drawn in this manuscript.